# Youth's exposure to and engagement with e-cigarette marketing on social media: a UK focus group study

Marissa J Smith ⬤ , Shona Hilton ⬤

## ABSTRACT

**Objective** Electronic-cigarettes (e-cigarette) are promoted creatively through social media and considering the potential influence of social media marketing on young people, we explored young people's exposure to and engagement with social media marketing of e-cigarettes.

**Design** Semistructured discussion groups.

**Subjects** Twenty focus groups with 82 young people aged 11–16 living in the Central belt of Scotland.

**Methods** Youths were asked about smoking and vaping behaviours, social media use, vaping advertisement exposure and were shown illustrative examples of social media content (eg, images and videos) about different messages, presentations and contextual features. Transcripts were imported into NVivo V.12, coded thematically and analysed.

**Results** Youths highlighted a variety of tactics e-cigarette companies use, including influencer or celebrity endorsement, attractive youth flavours, bright colours and emotional appeal to advertise and promote their products directly to young people. Social media influencers who advertise e-cigarettes were described as portraying e-cigarettes as 'cool' and 'fashionable' to entice viewers to try the products. Youths considered that there is a need for more restrictions on social media content to protect youths while also still allowing smokers to purchase them as a cessation device.

**Conclusions** Our study highlights that the e-cigarette industry is using previously employed tactics similar to the tobacco industry to advertise and promote its products on social media. These findings suggest the growing need for governments to work together to develop and implement policies to restrict the advertising and marketing of e-cigarettes on social media.

MRC/CSO Social and Public Health Sciences Unit, University of Glasgow, Glasgow, UK

**Correspondence to**
Marissa J Smith;
marissa.smith@glasgow.ac.uk

## STRENGTHS AND LIMITATIONS OF THIS STUDY

⇒ This study is one of the first in-depth studies to explore youths' responses to and engagement with electronic cigarette (e-cigarette) marketing on social media with a focus on the rise in user-generated and influencer content.

⇒ This research is very timely as governments consider the need for further e-cigarette regulations.

⇒ A methodological strength is the rich data set from semistructured interviews with 82 youths aged 11–16, facilitating understandings of youths' exposure to and interaction with e-cigarette marketing on social media.

⇒ Qualitative thematic analysis of the data allows depth of opinions but cannot offer predictions about the frequency of specific opinions with a wider population.

## INTRODUCTION

Marketing strategies used by electronic cigarette (e-cigarette) companies have been associated with the uptick in e-cigarette use (vaping) among youth.[1] Many studies have found that the promotion of e-cigarettes through various channels (eg, television (TV) and social media influencers) has lent to increased positive perceptions of vaping and intentions to use vaping products and contributed to e-cigarette uptake among youth.[2–4]

In the UK, the advertising of e-cigarettes is regulated through Article 20(5) of the revised European Union (EU) Tobacco Products Directive (2014/40/EU) (TPD),[5] which was transposed into UK law by the Tobacco and Related Products Regulations (TRPR) 2016.[6] The TPD prohibited the advertising of nicotine-containing e-cigarettes (unless licensed as medicines) in channels with potential cross-border impact (ie, channels that show adverts or sponsored events that originate from non-EU countries in EU countries), including TV, radio, newspapers, magazines and sponsorship.[7] Online advertising was also prohibited under the TPD, although the regulations left scope for marketers to retain websites containing factual information about e-cigarette products. As there are currently no medicinally licensed nicotine vaping products in the UK, the prohibitions apply to all nicotine-containing e-cigarette products on the market. The TRPR requirements for e-cigarette advertising were set out in 2017 and are enforced by the Committee of Advertising Practice (CAP)—a self-regulatory body of organisations representing advertising, direct marketing, media businesses

and sales promotion endorsed and administered by the independent Advertising Standards Authority (ASA).[8 9] Although Rule 22.12 of the ASA CAP code prohibits advertising in online media, social media content for e-cigarettes is permitted in 'non-paid-for space online under the marketer's control, provided claims are factual rather than promotional'.[10] Promotional content includes descriptive language that goes beyond objective, factual claims, significant imagery that is not related to the product, and health claims (eg, that e-cigarettes are safer or healthier than tobacco).[10]

Social media marketing includes promotions by means of paid digital advertising, via compensated 'influencers' (individuals with large followings who are paid to advertise brands and products), and on brand pages that companies can create for free and use to share content.[11] Most adolescents use visual-based social media daily. User-generated and influencer marketing content on social media represents a key influence on young people's understanding of products, including e-cigarettes.[12] E-cigarettes are promoted creatively through social media, with well-designed features including colours, flavour variations, emotional appeal, incentives (such as price promotions and discount vouchers) and even celebrity endorsements.[13 14] Research has shown that online marketing that leads to exposure to e-cigarette advertising, including where it is concealed as information or recommendations from peers, can increase the likelihood of vaping in individuals, including among youths.[15–17] In addition, a number of researchers have suggested that e-cigarette advertising could lead to increases in how socially acceptable and desirable cigarettes are perceived to be, and subsequently influence their continued use or possible (re)uptake in smokers, e-cigarette users and dual-users.[18–21] E-cigarette companies have been promoting their products on social media in an attempt to expand their market.[22 23] In addition, e-cigarette companies use social media platforms to promote e-cigarettes and create a socially attractive vaping image.[24] Previous studies (including references 25–27) have explored youths' perceptions of e-cigarette marketing on social media, finding that social media adverts entice non-smokers to use e-cigarettes, emotional appeals are featured in adverts, use of vibrant colours in the packaging, appeal of flavours and advertisements portray e-cigarettes as less harmful than combustible cigarettes. Liu et al[28] examined youths; perception of tobacco and marijuana messaging on social media and reported similar findings (including appealing colours in the packaging and appealing flavours). One limitation of the literature in this area is that it focuses on either smokers, non-smokers and/or vapers. This leaves out non-vapers and dual-users (those who use both cigarettes and e-cigarettes), understudied. It is important to establish whether exposure to e-cigarette adverts on social media influence attitudes towards e-cigarettes in non-vapers and smoking in dual-users.

Gaining a better understanding of the nature, extent and impact of e-cigarette marketing including their possible effects on youth e-cigarette perceptions and use can help guide the development of policies and interventions to educate youths about e-cigarettes.[29] This study, therefore, aims to explore youths', including non-smokers and non-vapers and dual-users exposure to, and engagement with e-cigarette advertising on social media, including user-generated and influencer content.

## METHODS

This section closely follows the methodological approach detailed in reference.[30] The data set has been rigorously analysed to answer two distinct research questions which have been answered in two separate manuscripts. Smith et al[30] explored youth's perceptions and engagement with e-cigarettes focusing on one type/generation of e-cigarettes; disposables. This manuscript explores young people's exposure to, and engagement with all types/generations of e-cigarettes but focuses on advertising on social media, including user-generated and influencer content.

We conducted 20 focus groups in Scotland between March and May 2022. Focus groups included between three and five participants (a total of 82 participants). Purposive sampling was used to recruit a diverse sample of youths in terms of age, sex, socioeconomic background, smoking status and vaping status.[31] Eleven groups were recruited through youth workers in local youth organisations. These gatekeepers handed out information sheets and helped achieve the sampling frame in terms of youth demographics and experiences concerning combustible cigarettes and e-cigarettes. The three organisations that helped with participant recruitment worked specifically with young people from disadvantaged backgrounds in urban areas. This recruitment strategy resulted in the inclusion of a range of participants from more and less deprived backgrounds and with experiences of smoking and vaping. Seven groups were recruited through the *Schools Health and Wellbeing Improvement Research Network (SHINE)* Newsletter which is distributed monthly to over 500 schools in Scotland. The remaining two groups were recruited via personal networks directly by MJS.

Focus group discussions were facilitated to allow the research team to explore opinions on online advertising and marketing of e-cigarettes. Friendship groups of 3–5 participants were used to facilitate in-depth insights and promote participant interaction. Each participant was given a £20 shopping voucher as compensation for their time.

Prior to the start of the focus groups, participants completed a short anonymous questionnaire about their age, biological sex, postcode, smoking and e-cigarette use status. For both combustible cigarettes and e-cigarettes, the questionnaire asked participants to specify whether they had tried or used them in the past or were using them at the time of the study. Based on a scoping review of user-generated content and influencer marketing involving e-cigarettes on social media,[14] a

**Table 1** Focus group location, participants and their cigarette smoking and e-cigarette use

| Group | Area-level deprivation | Sex | Age | Cigarette smoker | E-cigarette use |
|---|---|---|---|---|---|
| 1 | Least deprived | Female | 13–15 | Never | Never |
| 2 | Least deprived | Female | 14–15 | Never | Mixed—never (4)/tried (1) |
| 3 | Least deprived | Female | 13–16 | Never | Mixed—never (2)/tried (1) |
| 4 | Most deprived | Mixed—male (3)/female (2) | 12–15 | Mixed—never (3)/current (2) | Mixed—never (3)/tried (1)/current (1) |
| 5 | Most deprived | Mixed—male (1)/female (4) | 14–16 | Mixed—never (2)/tried (2)/current (1) | Mixed—never (2)/tried (2)/current (1) |
| 6 | Most deprived | Male | 12–15 | Never | Never |
| 7 | Most deprived | Male | 16 | Current | Current |
| 8 | Least deprived | Mixed—male (2)/female (3) | 14 | Never | Never |
| 9 | Most deprived | Male | 16 | Mixed—tried (1)/current (2) | Current |
| 10 | Most deprived | Mixed—male (4)/female (1) | 14–15 | Mixed—never (3)/tried (1)/current (1) | Mixed—never (3)/tried (1)/current (1) |
| 11 | Most deprived | Mixed—male (3)/female (2) | 13–16 | Mixed—never (2)/tried (2)/current (1) | Mixed—never (1)/current (4) |
| 12 | Least deprived | Mixed—male (2)/female (1) | 15–16 | Tried | Mixed—tried (2)/current (1) |
| 13 | Least deprived | Female | 13–16 | Never | Never |
| 14 | Most deprived | Mixed—male (1)/female (3) | 11–12 | Never | Never |
| 15 | Most deprived | Mixed—male (3)/female (1) | 11–12 | Never | Never |
| 16 | Most deprived | Mixed—male (2)/female (2) | 11–12 | Never | Never |
| 17 | Most deprived | Female | 14–16 | Mixed—never (4)/tried (1) | Mixed—never (1)/tried (1)/current (3) |
| 18 | Most deprived | Male | 13–16 | Never | Never |
| 19 | Most deprived | Female | 14 | Never | Mixed—tried (2)/current (1) |
| 20 | Least deprived | Female | 15–16 | Never | Tried (3) |

topic guide (online supplemental appendix A) was developed which covered three key areas, including use of social media, perceptions of social media influencers versus user-generated e-cigarette content on social media and views on the marketing of e-cigarettes on social media.

Images of different types of e-cigarettes ('tanks', disposables and pod devices) and e-liquids posted by social media influencers and general users were used as conversation starters found during the scoping review part of this research.[14] Group discussions were facilitated by MJS (a young post PhD early career researcher and experienced qualitative researcher). Ten of the groups were conducted online using Microsoft Teams and 10 of the groups were conducted face-to-face. Of these, one of the groups was conducted on the youth organisation's premises, and the other nine were conducted at the school, with representatives of the youth organisation present. Groups lasted between 40 and 66 min. Field-notes reflecting on the focus group and individual issues discussed were written up for each group. All focus groups were audio recorded (with participants' permission) and transcribed verbatim.

## Analysis

Following Braun and Clarke's six-phase framework for thematic analysis,[32] we conducted thematic analysis of the data from the focus groups. The research team read and reread the transcripts to become familiar with the data, and then iteratively constructed a coding frame to enable consistent organisation of relevant data. NVivo was used to organise categories on the basis of inductive themes that emerged from close reading of the, capture of both areas of agreement and less typical perspectives across a range of categories. Each transcript was imported into NVivo V.12, coded independently, cross-checked and analysed by MJS and SH (professor of public health and experienced qualitative researcher). Contradictory cases and group dynamics were discussed, making use of transcripts and field notes. The researchers reflected on their role as researcher, remained constantly aware of their position and took care not to introduce bias throughout the research. To further reduce bias, the researcher (MJS) recorded the focus groups and analysed them some time after they were completed ensuring a more reflective viewpoint of occurrences.

**Table 2**  E-cigarette use according to cigarette smoking

| Cigarette smoker | E-cigarette use | | | | | | | | | | | |
|---|---|---|---|---|---|---|---|---|---|---|---|---|
| | Never | | | Tried | | | Current | | | Total | | |
| | n | (col. %) | (row %) | n | (col. %) | (row %) | n | (col. %) | (row %) | n | (col. %) | (row %) |
| Never | 51 | 98.1 | 82.3 | 7 | 58.3 | 11.3 | 4 | 22.2 | 6.5 | 62 | 75.6 | 100.0 |
| Tried | 1 | 1.9 | 10.0 | 4 | 33.3 | 40.0 | 5 | 27.8 | 50.0 | 10 | 12.2 | 100.0 |
| Current | 0 | 0.0 | 0.0 | 1 | 8.3 | 10.0 | 9 | 50.0 | 90.0 | 10 | 12.2 | 100.0 |
| Total | 52 | 1 | 92.3 | 12 | 100.0 | 14.6 | 18 | 100.0 | 22.0 | 82 | 100.0 | 100.0 |

col., column.

## Patient and public involvement

Patients and/or the public were not involved in the design of the study, development of the research questions, recruitment, outcome measures or conduct of the study. A summary of the results will be disseminated to study participants.

## RESULTS
### Participant characteristics

Eighty-two youths aged 11–16 years participated (47 females (57%) and 35 males (43%)), representing diversity in sociodemographic characteristics and smoking-related behaviours. Age distribution within the sample was skewed slightly towards 14–15-year-olds, with 14-year-olds making up the largest subgroup (n=24). While the majority of participants did not currently smoke or use e-cigarettes, the sample included 10 smokers and 18 youths who used e-cigarettes. Deprivation rank was assigned using the Scottish Index of Multiple Deprivation (SIMD).[33] The area-level deprivation grouped using a binary deprivation variable (least deprived/most deprived) in which the three most deprived quintiles were grouped into the most deprived category for the Central Belt of Scotland area. Table 1 describes the focus group composition and participants in more detail and table 2 summarises smoking and e-cigarette use among the sample.

### Perceptions of advertising tactics used by the e-cigarette industry to target youths

Youths discussed three ways (influencer or celebrity endorsement, flavourings and emotional appeal) e-cigarette companies market their products to target young people.

### Influencer or celebrity endorsements

During the focus groups, participants frequently discussed the prevalence of social media influencers advertising e-cigarette products. When discussing why e-cigarette and/or tobacco companies would want to use social media influencers to advertise their products, participants highlighted the popularity of social media and how this would increase the reach of the adverts.

> I guess companies would use them [influencers] because the company may have approached them to advertise the product and they know they might have a huge following on the platform, so they might influence some more people to buy the products. (Male, current smoker, current vaper)

> More people are on social media nowadays. So, it's better that they [vaping/tobacco companies] use influencers to advertise their products compared to say in newspapers, 'cause you don't see people reading newspapers anymore. (Male, never smoker, never vaper)

Participants reported seeing marketing and advertising of e-cigarettes on social media and that they were portrayed as 'cool' and 'fashionable' by social media influencers.

> If an influencer posts it, a lot of people see it and be like, that's the trend now, it's a cool thing to do, so I want to take part in that. (Female, never smoker, tried vaping)

> It's like they're showcasing them. It's not just like it's in the background or whatever, it's not a normal pose to have them in, it's like they're showcasing them and that they are a fashionable thing to carry or to have. (Female, never smoker, never vaper)

Participants discussed that this would influence the viewer to try the product, thus increasing the prevalence of use.

> People get tempted easily by seeing the post. Like if they saw some influencers trying that stuff [e-cigarettes], people get tempted quickly to try that stuff, because then that person is trying it so the like the kids would think this is something that I should also try that stuff like that, that's the point. (Male, current smoker, current vaper)

> When you see what an influencer is, most people think oh if they're doing that, then it's cool, so it like, it's making more people want to buy the vapes, and stuff. (Female, never smoker, tried vaping)

Several participants discussed how social media posts make it too easy for underage youths to purchase and hide the fact you are buying e-cigarette products from websites.

I've seen a thing on TikTok like, they [influencers] show you them putting them [e-cigarettes] in the wee boxes and all that, or you could put them in secret packaging like behind the lashes. Like you can order it off their website and they'll hide it in the packaging, they put a few bits of sweeties on top of your vapes so your mum doesn't see it. (Female, tried smoking, current vaper)

### Using sweet and fruity flavours to appeal to youths

Participants discussed the variety of flavours of e-cigarettes available, highlighting that the sweet and fruity flavours are particularly attractive to youths.

'Cause younger people, like us would think that the flavoured ones look nice so they would try then and then they'd start vaping. (Male, never smoker, never vaper)

I think the type of ice cream flavours or sweetie ones are targeted at younger people because most people in their 30s and 40s would probably use the tobacco one or the coffee ones. (Male, current smoker, current vaper)

The ones that stood out to me the most were the Ben and Jerry's and fruit ones compared to the tobacco ones. I feel like, if I was going to buy them, I would buy them 'cause they are attractive and are things I like. (Female, tried smoking, current vaper)

Several participants stated that the variety of flavourings was to appeal to youths.

I feel like no one really wants to taste tobacco, to be honest. But I think putting out there with like grape and blueberry flavourings are going to attract younger people to the marketplace. I think it's an intentional design and I feel like that is probably what it does attract youths. (Female, never-smoker, never vaper)

### USING VIBRANT COLOURS TO ATTRACT YOUTH ATTENTION

In addition, participants spoke about the presentation of e-cigarette products on social media, for example, the use of vibrant colours and filters.

You can tell by the picture they're promoting them. Because if it was not promoting them you would have like a red colour, it's like a stop, it's kind of like a no colour. But they are using really positive colours, like pink and green. It makes you stop and look at it to see what it's [the social media post] is about. (Male, never smoker, never vaper)

It was really bright colours and they had bright filters on them so it is brighter. If you were scrolling past then gives you a bright colour, you'd go back and see it. You'd be like, oh what's that? You'd be, like, drawn to it. […] I think young people like us would be drawn to that, well I would. (Female, never smoker, never vaper)

### Connecting emotionally with youths

Participants discussed the emotional connections and positive portrayal of e-cigarettes on social media, stating that individuals in the social media posts 'looked happy' to be using the products.

I think the person who posted this product, I think he or she was trying to emphasise the fact that these products are good because they look happy to be advertising it. So, you should try them and like they were not promoting vaping negatively or that it is bad at all. I think they were trying to sell the aesthetic idea of this product. (Male, current smoker, current vaper)

It's the way they're all holding them, they looked quite passionate with a smile on their face. A big cheesy smile. (Male, current smoker, current vaper)

Participants highlighted the use of emojis in e-cigarette social media posts. Emojis (icons used to express an idea or emotion) and emoji sequencing (the use of two or more emojis that form a conceptual rhetorical unit)[34] are used to enhance the message meaning.[35 36] When asked if this would affect their interaction with social media posts that included emojis, participants stated that they would be more inclined to read the post, rather than scrolling past.

If you see it in colours, like emojis, you're like, oh I wonder what that is. You'd stop scrolling. 'Cause normally captions is just, like, text and a black background or something like that. (Female, never smoker, never vaper)

The wee emojis they would draw my attention, because of how brightly coloured they are. And I think that's what they're trying to do, like, it'll draw your attention. (Female, never smoker, never vaper)

### Perceptions of what can be done to limit e-cigarette marketing on social media

Participants had divergent views on the marketing and advertising of e-cigarettes on social media. Several participants stated that marketing and advertising should not be allowed as youths, including those of a similar age, could access this type of content and it may encourage them to try e-cigarettes.

Advertising them [e-cigarettes] on social media is s a bad thing because younger people might see it and then try it. (Male, never smoker, never vaper)

I don't think it [advertising of e-cigarettes] should be allowed, just because of the younger community on it [social media]. And then if they are seeing them, they might end up doing it, getting addicted and end up dying. (Male, tried smoking, tried vaping)

But participants also acknowledged that controlling what is posted on social media is 'tricky' and 'difficult'

and that prohibiting e-cigarette advertising may limit their role as cessation devices by smokers. Those participants suggested instead of prohibiting the marketing and advertising of e-cigarettes, warnings (such as age and health warnings) or other restrictive measures (such as parental controls or age restrictions) could be imposed on these types of social media posts.

> I think there needs to be age restrictions on the posts or parents could control what their child sees. (Female, never smoker, never vaper)

## DISCUSSION

E-cigarettes have become increasingly popular and visible in public life, particularly on social media. Our qualitative thematic analysis of youths' perceptions of and engagement with e-cigarette advertising on social media highlighted a public health concern. Similar to previous research,[37–39] youths in our study highlighted the tactics used by the e-cigarette industry to promote and advertise e-cigarettes to underage populations through influencer and celebrity endorsements, variety of flavours and using promotional endorsements. Youths discussed the positive portrayal of e-cigarettes by social media influencers, and they were enticing the viewer to buy the product. Youths highlighted that e-cigarette and/or tobacco companies use social media influencers to advertise their products on social media and by adverting on social media platforms, they will increase the reach of the adverts to a wider audience. We found that youths had divergent views on e-cigarette marketing on social media. Several participants stated that e-cigarette advertising on social media should be prohibited, citing the high prevalence of youths on social media who could access the content. Whereas others suggested a balanced approach to regulating advertising on social media so that they can still be advertised as cessation devices by smokers. Youths suggested that all adverts should contain warnings (such as age and health warnings) or other restrictive measures (such as parental controls or age restrictions) to protect youths and non-users.

Previous studies which examined e-cigarette-related social media[25–27 40–44] found that the vast majority of the content depicted positive attitudes towards vaping, while negative characterisations were mostly absent. Our study found that youths had similar experiences when exposed to e-cigarette content on social media. Consequently, there is a risk that youths will be exposed to, and possibly engage with, content that promotes vaping while staying uninformed about the negative aspects, including potential health harms. Our study adds to existing concerns about youths' perceptions of e-cigarettes as cool, fashionable products,[30 45–48] by considering the advertising of the products by influencers on social media platforms, frequently accessed by youths. The use of social media influencers to promote products and celebrity endorsements encourage their followers to share and interact with their content, and ultimately purchase their products. Social media influencers often collaborate with multiple industries (eg, fashion and beauty products) in addition to e-cigarette products/brands. These influencers could potentially expose their non-e-cigarette-focused audience (including non-users of e-cigarettes) to e-cigarette content.[49] Therefore, these influencers could be considered an even higher risk for youth compared with influencers who post exclusively about e-cigarettes. This is problematic and concerning as most e-cigarette brands claim that their advertising and promotional content is meant to target current cigarette smokers to help them switch to e-cigarettes,[50 51] rather than people who do not use nicotine at all. Previous literature has focused on either smokers, non-smokers and/or vapers, leaving out non-vapers and dual-users. Participants in our study were primarily non-smokers and non-vapers, we argue that it is important to explore non-smokers and non-vapers exposure to and engagement with e-cigarette adverts on social media. This is important for helping to determine the potential influence of e-cigarette adverts on social media in non-smokers and non-vapers.

Before plain packaging, the tobacco industry successfully used colourful packaging to represent its brand and increase the appeal of smoking.[52–54] The marketing tactics used by the tobacco industry were so effective that several e-cigarette manufacturers have used the same trade secrets to advertise their products.[55–57] Research has shown the plethora of bright-coloured packaging, eye-catching designs and variety of flavourings available all appeal to youths and may result in experimentation of the products.[58–63]

Previous research has illustrated YouTube viewers' exposure to e-cigarettes (as well as alcohol and tobacco) while watching videos on the platform.[64] Our study adds to this by illustrating the potential mechanisms by which exposure to e-cigarette advertising on social media platforms may influence perceptions and resultant trial/use of the products. While restrictions on mass media marketing of e-cigarettes are increasingly considered internationally, social media platforms are more difficult to manage because user-generated content (in particular) will not be covered by incoming regulations. Our study findings resonate with previous research, which found that youths were easily able to purchase e-cigarettes from the internet due to the absence of age-verification measures used by internet vendors.[65 66]

As with all research, our study has some limitations, several of which are similar to that detailed in reference[30] as this research closely followed the same methodological approach. First, and consistent with the qualitative design, the sample does not aim to be representative of UK youth, as our study focused on Scottish youths. However, we did have a diverse sample of both sexes. Second, the study's geographical remit has to be considered when interpreting the findings. The UK is considered as an international leader in tobacco control policy. It is possible that participants' views may have been influenced by

the UK's unique favourable policy approach to e-cigarettes and legal and sociocultural context, including low smoking prevalence. Third, using friendship groups may have influenced youth's honesty when answering questions and their responses. Youths may not have wanted to disagree with their peers, thus responded in a similar way to the rest of the group. Fourth, participants in this study were primarily non-smokers and non-vapers and it is possible that this may have impacted in their discussion of e-cigarette advertisements on social media. However, as we used predominately visual prompts based on data found during the scoping review of the research,[14] we did not find that non-smokers and non-vapers contributed less to discussions. If we had not used visual prompts and focused on participants own experiences of what they had seen on social media (ie, recalling from memory), this may have impacted non-smokers and non-vapers contribution to discussions. Fifth, the data were collected in different formats (online and face-to-face), and this may have influenced participants' responses. Two of the online groups were conducted in a classroom with a teacher present, and during seven face-to-face groups, a teacher/youth worker was present in the room. The presence of a teacher/youth worker may have influenced youth's willingness to answer questions and their responses. Finally, two of the groups were recruited through personal networks and this may have impacted the youth's responses.

The results from this study should provide guidance for future research. More research is needed to determine the most effective means to counter the favourable/positive aspects of e-cigarettes to reduce youths' interest in product trial and use. Future research could explore if there are different perceptions of e-cigarette exposure in social media between females and males. As females are one of the target groups for e-cigarette use,[67 68] it can reflect the marketing situation of e-cigarettes for females on social media, which deserves further study. In addition, research on the prevalence of warning statements on e-cigarette-related social media posts and the impact this has on youth perceptions would be an important future direction to expand this work.

Considering that most youth access multiple social media platforms multiple times per day[69 70] and that exposure to this marketing is related to use,[71 72] our findings indicate a significant public health concern. Given the findings presented by our study, there is a growing need for policymakers to develop comprehensive plans to build more robust measures to protect youths and to restrict the ability of marketers to reach youths with enticing social media content promoting e-cigarettes. In addition, social media platforms should consider implementing more robust measures, such as age restrictions and portraying the negative aspects of vaping, to ensure the prevention of vaping-related content targeted at underage users.

## CONCLUSION

The tactics used on social media platforms for advertising and marketing e-cigarettes directly to youths raise concerns and a new generation of youths becoming addicted to nicotine. The results of this study highlight that the e-cigarette industry is using previously employed tactics similar to the tobacco industry to advertise and promote its products on social media. Most youths use social media daily and the depiction of products on social media represents a key influence on young people's understanding of products. These findings may highlight a priority for governmental policy to restrict the ability of marketers to reach youths with social media content promoting e-cigarettes.

**Contributors** MJS: guarantor, conceptualisation, data curation, investigation, methodology, validation, visualisation, writing—original draft preparation. SH: conceptualisation, methodology, validation, writing—review and editing.

**Funding** MS acknowledges funding from Cancer Research UK grant PPRCTAGPJT\100003. SH is funded by the Medical Research Council grant MC_UU_00022/1, the Chief Scientist Office of the Scottish Government Health Directorates grant SPHSU17, and Cancer Research UK grant PPRCTAGPJT\100003.

**Competing interests** None declared.

**Patient and public involvement** Patients and/or the public were not involved in the design, or conduct, or reporting or dissemination plans of this research.

**Patient consent for publication** Consent obtained directly from patient(s).

**Ethics approval** This study involves human participants. Ethical approval for the study was obtained from the University of Glasgow's Medical and Veterinary Life Sciences Ethics Committee (reference 200210034). Participants gave informed consent to participate in the study before taking part.

**Provenance and peer review** Not commissioned; externally peer reviewed.

**Data availability statement** All data relevant to the study are included in the article or uploaded as supplementary information.

**ORCID iDs**
Marissa J Smith http://orcid.org/0000-0002-5017-6085
Shona Hilton http://orcid.org/0000-0003-0633-8152

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
