## [Reviewer comments · BMJ Open]

ARTICLE DETAILS

TITLE (PROVISIONAL)	Youth's exposure to and engagement with e-cigarette marketing on social media: a UK focus group study
AUTHORS	Smith, Marissa; Hilton, Shona

VERSION 1 – REVIEW

REVIEWER	McQuoid, Julia The University of Oklahoma, TSET Health Promotion Research Center
REVIEW RETURNED	25-Jan-2023

GENERAL COMMENTS	Thank you for the opportunity to review this manuscript, which describes a qualitative study of youths' engagement with and perspectives on e-cigarette marketing on social media. The authors should be commended for the substantial amount of work that must have gone into the number of focus groups conducted and the volume of data that was analyzed. Please see my suggestions below for the paper. All my best to the authors in their future endeavors. TITLE I suggest editing the title to be a little more specific and informative so that the reader can have an idea about what the article contributes on this topic of youth exposure to e-cigarette marketing on social media. What was the main contribution of the study to the literature? There may be a way to highlight that in the title. GENERAL Please check for grammatical errors and typos throughout the paper. For example, in "youth's responses to, and engagement with e-cigarette advertising on social media", the apostrophe is in the wrong place in "youths" and I believe the sentence is missing a comma after "engagement with". There are several other errors like this to catch throughout the paper. If helpful, I wanted to suggest this article for reference. I find it helpful for enhancing the quality of qualitative research papers: Tracy, S. J. (2010). Qualitative Quality: Eight "Big-Tent" Criteria for Excellent Qualitative Research. Qualitative Inquiry, 16(10), 837–851. https://doi.org/10.1177/1077800410383121 ABSTRACT -In the Objective, please identify a more specific research gap that will be explored within the context of prior research on adolescents' perceptions of social media marketing of e-cigarettes. STRENGTHS AND LIMITATIONS
---

-First bullet point – there appears to be a typo in this sentence.

-Third bullet point – I don't understand this sentence. Do the authors mean 'qualitative thematic analysis'? Also, I wonder if the authors are referring to generalizability rather than ability to predict forward in time.

-Fourth bullet point – this sentence appears to contain two separate ideas. Please separate out for clarity.

INTRODUCTION

-The introduction would be strengthened by a more thorough review of what is already known about the stated topic of this paper: youths' responses to, and engagement with, e-cigarette advertising on social media. I thinking honing in on and describing findings from the existing studies on this topic will help to identify a more specific research gap for this paper. What is already known about youth's responses to, and engagement with e-cigarette advertising on social media (for example, what themes have been identified in qualitative studies?). What is unknown that this study was designed to find out? Here are a few articles in case they're helpful in reviewing the literature and integrating it into the introduction:

Liu J, McLaughlin S, Lazaro A, Halpern-Felsher B. What Does It Meme? A Qualitative Analysis of Adolescents' Perceptions of Tobacco and Marijuana Messaging. *Public Health Reports*. 2020;135(5):578-586. doi:10.1177/0033354920947399

Andrea C. Johnson, Darren Mays, Kirsten B. Hawkins, Molly Denzel & Kenneth P. Tercyak (2017) A qualitative study of adolescent perceptions of electronic cigarettes and their marketing: Implications for prevention and policy, *Children's Health Care*, 46:4, 379-392, DOI: 10.1080/02739615.2016.1227937

Yvonne Chen, Chris Tilden & Dee Katherine Vernberg (2020) Adolescents' interpretations of e-cigarette advertising and their engagement with e-cigarette information: results from five focus groups, *Psychology & Health*, 35:2, 163-176, DOI: 10.1080/08870446.2019.1652752

Scheffels, J., et al. (2023). "E-cigarette use in global digital youth culture. A qualitative study of the social practices and meaning of vaping among 15–20-year-olds in Denmark, Finland, and Norway." *International Journal of Drug Policy* 111: 103928.

-I think the paper would be more accurately framed as a study of engagement with and perceptions of social media e-cigarette marketing with primarily non-vaping/non-smoking youths given the composition of the sample. Please adjust the introduction to reflect this and to make an argument for why a study with this population of primarily non-smokers/non-vapers on this particular topic is significant and worthy of study.

METHODS

-It is unclear why such a large number of focus groups and such a large sample size was necessary to achieve the sample composition necessary to answer the research question of the study. Please elaborate on how sampling and saturation proceeded.

-Were participants purposively recruited for age?

-Unless participants were asked about biological sex or sex assigned at birth, I suggest using the term 'gender' rather than 'sex'.

-Relatedly, were participants offered any other gender identity constructs to choose from other than 'male' and 'female'? In a large sample of young people it seems unlikely that there wouldn't be greater gender diversity (e.g., gender non-binary, gender queer, transgender, etc).

-It is stated that participants were purposively sampled for smoking status and e-cigarette status, but the sample was overwhelmingly comprised of young people who did not use combustible cigarettes or e-cigarettes. Please explain how this happened. Especially given the large sample I would have expected that greater diversity in smoking/vaping status could have been achieved if that was a sampling goal.

-Line 101 – I suggest considering terminology like 'privileged' and 'disadvantaged' rather than affluent and deprived.

-Line 105 – typo in this sentence

-How were friendship groups determined? Also, are there any limitations to grouping participants by friendship groups in the focus group discussion?

-Line 110 – I suggest using the term 'combustible cigarettes' rather than 'traditional cigarettes'

-Line 112-113 states that the interview guide was developed with a review of the literature. Please provide the citations for the literature that informed the development of the interview guide.

-Please provide a more detailed description of what was in the interview guide and how questions were posed to the focus group participants.

-Line 116 – how were the images shown to participants selected?

-To enhance the 'sincerity' of the article, please provide information about the authors (e.g., age relative to the participants, education, position, training in qualitative research methods, training in focus group facilitation)

-Please make the description of data analysis its own paragraph(s) in the methods section and describe the process of analysis in more detail. What kind of analytic approach was taken? Was there a codebook? If so, how was it developed?

FINDINGS

-Table 1 – To my reading, the description of residential areas as either 'affluent' or 'deprived' is problematic. Why was this binary approach taken to describing residential areas and how was this decision arrived at? Is there a more nuanced way to describe the characteristics and composition of these residential areas, perhaps from a different information source?

-The findings section is logically organized and the quotes support the themes that are presented.

	DISCUSSION -What does this study contribute that was not already in the literature? As noted in my comments about the introduction, a more thorough review of the existing literature on this topic in the introduction could identify a more specific research gap to motivate the study, and this literature could be brought through to the discussion to offer a more nuanced discussion of what novel findings this study contributes to the existing body of knowledge. -Please acknowledge that these youth were primarily non-smokers/non-vapers and discuss how this may have impacted their discussion of e-cigarette marketing on social media. Please also discuss why it is important to understand this primarily non-smoker/non-vaper group of youths' perspective on and experiences with this topic.
--	---

REVIEWER	Zhu, Jingfen Shanghai Jiao Tong University, School of Public Health
REVIEW RETURNED	02-Mar-2023

GENERAL COMMENTS	The study explored young people's perceptions of social media marketing of e-cigarettes through semi-structured discussion groups and found some worrying result. The findings urged restriction of e-cigarette's advertising and marketing on social media in Scotland. But there are some suggestion as follows I don't understand whether this study analyzed the 28 adolescents who had used tobacco products or all participants . In fact, the perceptions of social media marketing among non-users is also meaningful and can be compared with smokers or e-cigarettes users. In addition, the study included more girls than boys, but it is not clear whether there were different perceptions of e-cigarette exposure in social media between boys and girls in the results. As female are one of the target groups for e-cigarette use, it can reflect the marketing situation of e-cigarettes for female on social media ,which deserves detailed discussion.
--

VERSION 1 – AUTHOR RESPONSE

Reviewer: 1

Thank you for the opportunity to review this manuscript, which describes a qualitative study of youths' engagement with and perspectives on e-cigarette marketing on social media. The authors should be commended for the substantial amount of work that must have gone into the number of focus groups conducted and the volume of data that was analysed. Please see my suggestions below for the paper. All my best to the authors in their future endeavours.

- We thank the reviewer for their helpful comments, please see below for a point-to-point response to each comment.

R1.1 I suggest editing the title to be a little more specific and informative so that the reader can have an idea about what the article contributes on this topic of youth exposure to e-cigarette marketing on

social media. What was the main contribution of the study to the literature? There may be a way to highlight that in the title.

- We thank the reviewer for their comment and in line with the BMJ Open guidelines we have revised the title as follows:

“Youth’s exposure to and engagement with e-cigarette marketing on social media: a UK focus group study”

R1.2 Please check for grammatical errors and typos throughout the paper. For example, in “youth’s responses to, and engagement with e-cigarette advertising on social media”, the apostrophe is in the wrong place in “youths” and I believe the sentence is missing a comma after “engagement with”. There are several other errors like this to catch throughout the paper.

- We thank the reviewer for their comment, we have reviewed the manuscript for any grammatical errors and typos.

R1.3 If helpful, I wanted to suggest this article for reference. I find it helpful for enhancing the quality of qualitative research papers:

Tracy, S. J. (2010). Qualitative Quality: Eight “Big-Tent” Criteria for Excellent Qualitative Research. *Qualitative Inquiry*, 16(10), 837–851. <https://doi.org/10.1177/1077800410383121>

- We thank the reviewer for highlighting this paper, we have revised the manuscript to include more methodological details, similar to those in the suggested article. Please see the manuscript and our response to R1.22 for our amendments.

R1.4 In the Objective, please identify a more specific research gap that will be explored within the context of prior research on adolescents’ perceptions of social media marketing of e-cigarettes.

- We thank the reviewer for their comment, we have reviewed the objective as follows:

“Objective: E-cigarettes are promoted creatively through social media and considering the potential influence of social media marketing on young people, we explored young people’s exposure to and engagement with social media marketing of e-cigarettes.”

R1.5 First bullet point – there appears to be a typo in this sentence.

- We have revised the manuscript to fix this typo.

R1.6 Third bullet point – I don’t understand this sentence. Do the authors mean ‘qualitative thematic analysis’? Also, I wonder if the authors are referring to generalizability rather than ability to predict forward in time.

- We apologise for any confusion. In response to E1.3 we have revised the strengths and limitations section as follows:

“Strengths and limitation of this study

- This study is one of the first in-depth studies to explore youths' responses to and engagement with e-cigarette marketing on social media with a focus on the rise in user-generated and influencer content.
- This research is very timely as governments consider the need for further e-cigarette regulations.
- A methodological strength is the rich dataset from semi-structured interviews with 82 youths aged 11-16, facilitating understandings of youths' exposure to and interaction with e-cigarette marketing on social media.
- Qualitative thematic analysis of the data allows depth of opinions but cannot offer predictions about the frequency of specific opinions with a wider population."

R1.7 Fourth bullet point – this sentence appears to contain two separate ideas. Please separate out for clarity.

- In response to E1.3 we have revised the strengths and limitations section. Please see response to E1.3 or the tracked manuscript for our changes.

R1.8 The introduction would be strengthened by a more thorough review of what is already known about the stated topic of this paper: youths' responses to, and engagement with, e-cigarette advertising on social media. I thinking honing in on and describing findings from the existing studies on this topic will help to identify a more specific research gap for this paper. What is already known about youth's responses to, and engagement with e-cigarette advertising on social media (for example, what themes have been identified in qualitative studies?). What is unknown that this study was designed to find out? Here are a few articles in case they're helpful in reviewing the literature and integrating it into the introduction:

Liu J, McLaughlin S, Lazaro A, Halpern-Felsher B. What Does It Meme? A Qualitative Analysis of Adolescents' Perceptions of Tobacco and Marijuana Messaging. *Public Health Reports*. 2020;135(5):578-586. doi:10.1177/0033354920947399

Andrea C. Johnson, Darren Mays, Kirsten B. Hawkins, Molly Denzel & Kenneth P. Tercyak (2017) A qualitative study of adolescent perceptions of electronic cigarettes and their marketing: Implications for prevention and policy, *Children's Health Care*, 46:4, 379-392, DOI: 10.1080/02739615.2016.1227937

Yvonne Chen, Chris Tilden & Dee Katherine Vernberg (2020) Adolescents' interpretations of e-cigarette advertising and their engagement with e-cigarette information: results from five focus groups, *Psychology & Health*, 35:2, 163-176, DOI: 10.1080/08870446.2019.1652752

Scheffels, J., et al. (2023). "E-cigarette use in global digital youth culture. A qualitative study of the social practices and meaning of vaping among 15–20-year-olds in Denmark, Finland, and Norway." *International Journal of Drug Policy* 111: 103928.

- We thank the reviewer for their comment and for suggesting literature to review. Please see the introduction section of the manuscript for our tracked changes.

R1.9 I think the paper would be more accurately framed as a study of engagement with and perceptions of social media e-cigarette marketing with primarily non-vaping/non-smoking youths given the composition of the sample. Please adjust the introduction to reflect this and to make an argument for why a study with this population of primarily non-smokers/non-vapers on this particular topic is significant and worthy of study.

- We thank the reviewer for their comment, we have revised the title and introduction of the manuscript to frame the study as one which explores the engagement with and perceptions of social media e-cigarette marketing. Please see R1.1 for our revised title and the manuscript for amendments to the introduction.

R1.10 It is unclear why such a large number of focus groups and such a large sample size was necessary to achieve the sample composition necessary to answer the research question of the study. Please elaborate on how sampling and saturation proceeded.

- We thank the reviewer for their comment. We received ethical approval to recruit between 40-100 participants and this range was selected as we believed it would allow us to explore a range of experiences across the different regions in the Central belt of Scotland. We found that increasingly there was little new data emerging and by the 20th focus group considered we had reached 'saturation'. In addition, we wanted to ensure we had representation from youths across different area-level deprivation. Furthermore, we have revised the manuscript to include reference to GUEST, G., BUNCE, A. & JOHNSON, L. 2006. How Many Interviews Are Enough?: An Experiment with Data Saturation and Variability. *Field Methods*, 18, 59-82.

R1.11 Were participants purposively recruited for age?

- This study aimed to recruit youths aged 11-16 years old, thus participants were participants purposively recruited for age. We have revised the manuscript as follows to state this:

"Purposive sampling was used to recruit a diverse sample of youths in terms of age, sex, socio-economic background, smoking status, and vaping status."

R1.12 Unless participants were asked about biological sex or sex assigned at birth, I suggest using the term 'gender' rather than 'sex'.

- We thank the reviewer for their comment. We can confirm that we participants were asked about biological sex, hence our use of the term. We have revised the manuscript to indicate this. It is worth noting that based upon discussion within the researcher team and following extensive feedback from the ethics committee it was advised we use the term 'sex' rather than 'gender'. Given the age range of participants (11-16 years old) we were advised to use 'sex' as the younger youths may not be aware of gender identity constructs. It was highlighted to the research team that it was not our place to discuss/educate youths on gender constructs as was out with the scope of the research. While we do appreciate it is important consider and acknowledge gender identity constructs, on this instance we followed the feedback and advice from the University ethics committee.

R1.13 Relatedly, were participants offered any other gender identity constructs to choose from other than 'male' and 'female'? In a large sample of young people, it seems unlikely that there wouldn't be greater gender diversity (e.g., gender non-binary, gender queer, transgender, etc).

- Please see our response to R1.12

R1.14 It is stated that participants were purposively sampled for smoking status and e-cigarette status, but the sample was overwhelmingly comprised of young people who did not use combustible cigarettes or e-cigarettes. Please explain how this happened. Especially given the large sample I

would have expected that greater diversity in smoking/vaping status could have been achieved if that was a sampling goal.

- We thank the reviewer for their comment. We acknowledge that the sample predominately comprised of young people who did not use combustible cigarettes or e-cigarettes. Of the 82 participants in the sample 10 (12.2%) were smokers and 18 (22%) were vapers. Our purpose target was to include 10% of our total sample as smokers/vapers, thus our sample met these criteria.

R1.15 Line 101 – I suggest considering terminology like ‘privileged’ and ‘disadvantaged’ rather than affluent and deprived.

- We thank the reviewer for their comment. We have updated the terminology to most/least deprived area based deprivation category using Scottish Index of Multiple Deprivation (SIMD). We have revised the participant characteristics section (see below) and Table 1 (see manuscript) of the manuscript.

“Deprivation rank was assigned using the Scottish Index of Multiple Deprivation (SIMD) [1]. The area-level deprivation was grouped using a binary deprivation variable (least deprived/most deprived) in which the three most deprived quintiles were grouped into the most deprived category for the Central Belt of Scotland area.”

R1.16 Line 105 – typo in this sentence

- We have revised the manuscript to fix this typo.

R1.17 How were friendship groups determined? Also, are there any limitations to grouping participants by friendship groups in the focus group discussion?

- We thank the reviewer for their comment. Friendship groups were determined by the participants themselves or organised through the organisation which the participants were recruited via. We agree with the reviewer regarding discussing the limitations of friendship groups, we have revised the discussion as follows:

“Thirdly, using friendship groups may have influenced youth’s honesty when answering questions and their responses. Youth’s may not have wanted to disagree with their peers, thus responded in a similar way to the rest of the group.”

R1.18 Line 110 – I suggest using the term ‘combustible cigarettes’ rather than ‘traditional cigarettes’

- We thank the reviewer for their comment, we have changed the term traditional cigarettes’ to ‘combustible cigarettes’.

R1.19 Line 112-113 states that the interview guide was developed with a review of the literature. Please provide the citations for the literature that informed the development of the interview guide.

- We thank the reviewer for their comment, we have revised the manuscript as follows:

“Based on a scoping review of user-generated content and influencer marketing involving e-cigarettes on social media [2], a topic guide was developed which covered three key areas, including use of social media, perceptions of social media influencers versus user-generated e-cigarette content on social media, and views on the marketing of e-cigarettes on social media.”

R1.20 Please provide a more detailed description of what was in the interview guide and how questions were posed to the focus group participants.

- In response to the editor’s comment (E1.4), we have included a copy of the topic guide as supplementary material. It should be noted that the images used in the focus groups were shown in a PowerPoint presentation. A series of images (e.g., Slides 2-8) were shown and changed at five second intervals, once all images in that section were shown the group were then asked to discuss these based upon question in the topic guide. As it is not possible to replicate this in the manuscript, we have included copies of the images within the topic guide (Appendix A).

R1.21 Line 116 – how were the images shown to participants selected?

- We thank the reviewer for their comment, we have revised the manuscript as follows:

“Images of different types of e-cigarettes (‘tanks’, disposables, and pod devices) and e-liquids posted by social media influencers and general users found during the scoping review part of this research [2], were used as conversation starters.”

R1.22 To enhance the ‘sincerity’ of the article, please provide information about the authors (e.g., age relative to the participants, education, position, training in qualitative research methods, training in focus group facilitation)

- We thank the reviewer for their comment, we have revised the manuscript to include more provide information about the authors.

Line 150: Group discussions were facilitated by MS (a young post PhD early career researcher post and experienced qualitative researcher).

Line 165: Each transcript was imported into NVivo 12, coded independently, cross-checked, and analysed by MS and SH (professor of public health and experienced qualitative researcher).

Line 168: The researchers reflected on their role as researcher, remained constantly aware of their position and took care not to introduce bias throughout the research. To further reduce bias the researchers recorded the focus groups and analysed them some time after they were completed ensuring a more reflective viewpoint of occurrences.

R1.23 Please make the description of data analysis its own paragraph(s) in the methods section and describe the process of analysis in more detail. What kind of analytic approach was taken? Was there a codebook? If so, how was it developed?

- We agree with the reviewer that it would be beneficial to separate out the analysis stage and for more detail to be added. We have revised the manuscript as follows:

“Analysis

Following Braun and Clarke’s six-phase framework for thematic analysis [3], we conducted thematic analysis of the data from the focus groups. The research team read and re-read the transcripts to become familiar with the data, and then iteratively constructed a coding frame to enable consistent organisation of relevant data. NVivo was used to organise categories on the basis of inductive themes that emerged from close reading of the, capture of both areas of agreement and less typical perspectives across a range of categories. Contradictory cases and group dynamics were discussed, making use of transcripts and field notes. Ethical approval for the study was obtained from the University of Glasgow’s Medical and Veterinary Life Sciences Ethics Committee (reference 200210034).”

R1.24 Table 1 – To my reading, the description of residential areas as either ‘affluent’ or ‘deprived’ is problematic. Why was this binary approach taken to describing residential areas and how was this decision arrived at? Is there a more nuanced way to describe the characteristics and composition of these residential areas, perhaps from a different information source?

- Please see our response to R1.15.

R1.25 The findings section is logically organized and the quotes support the themes that are presented.

- We thank the reviewer for their comment.

R1.26 What does this study contribute that was not already in the literature? As noted in my comments about the introduction, a more thorough review of the existing literature on this topic in the introduction could identify a more specific research gap to motivate the study, and this literature could be brought through to the discussion to offer a more nuanced discussion of what novel findings this study contributes to the existing body of knowledge.

- We thank the reviewer for their comment, in response to this comment and R1.8 we have revised the introduction of the manuscript, please the manuscript for our tracked changes. In addition, we have revised the discussion section of the manuscript, please the manuscript for our tracked changes.

R1.27 Please acknowledge that these youth were primarily non-smokers/non-vapers and discuss how this may have impacted their discussion of e-cigarette marketing on social media. Please also discuss why it is important to understand this primarily non-smoker/non-vaper group of youths’ perspective on and experiences with this topic.

- We thank the reviewer for their comment. We have revised the discussion section of the manuscript to acknowledge that participants were primarily non-smokers and non-vapers and how this may have impacted discussions.

Line 389: Fourthly, participants in this study were primarily non-smokers and non-vapers and it is possible this may have impacted in their discussion of e-cigarette advertisements on social media. However, as we used predominately visual prompts based upon data found during the scoping review of the research [2] we did not find that non-smokers and non-vapers contributed less to discussions. If we had not used visual prompts and focused on participants own experiences of what they had seen

on social media (i.e., recalling from memory) this may have impacted non-smokers and non-vapers contribution to discussions.

- We have revised the manuscript (see below) to discuss why it is important to understand this primarily non-smoker/non-vaper group of youths' perspective on and experiences with this topic.

Line 110: One limitation of the literature in this area is that it focuses on either smokers, non-smokers and/or vapers. This leaves out non-vapers and dual-users (those who use both cigarettes and e-cigarettes), understudied. It is important to establish whether exposure to e-cigarette adverts on social media influence attitudes towards e-cigarettes in non-vapers and smoking in dual-users.

Line 359: Previous literature has focused on either smokers, non-smokers and/or vapers, leaving out non-vapers and dual-users. Participants in our study were primarily non-smokers and non-vapers, we argue that it is important to explore non-smokers and non-vapers exposure to and engagement with e-cigarette adverts on social media. This is important for helping to determine the potential influence of e-cigarette adverts on social media in non-smokers and non-vapers.”

Reviewer: 2

The study explored young people's perceptions of social media marketing of e-cigarettes through semi-structured discussion groups and found some worrying result. The findings urged restriction of e-cigarette's advertising and marketing on social media in Scotland. But there are some suggestions as follows:

R2.1 I don't understand whether this study analyzed the 28 adolescents who had used tobacco products or all participants. In fact, the perceptions of social media marketing among non-users is also meaningful and can be compared with smokers or e-cigarettes users.

- We thank the reviewer for their comment. To confirm, we conducted 20 focus groups with 82 youths (aged 11-16), of the 82, 62 had never used tobacco products, 10 have tried tobacco products and 10 are current users of tobacco products. We analysed opinions and responses from all 82 participants. We agree with the reviewer that the perceptions of social media marketing among non-users is also meaningful and can be compared with smokers or e-cigarettes users and we have revised the introduction and discussion sections of the manuscript to reflect this. Please the manuscript for our tracked changes.

R2.2 In addition, the study included more girls than boys, but it is not clear whether there were different perceptions of e-cigarette exposure in social media between boys and girls in the results. As female are one of the target groups for e-cigarette use, it can reflect the marketing situation of e-cigarettes for female on social media ,which deserves detailed discussion.

- We thank the reviewer for their comment. While our study focused exploring young people's exposure to and engagement with social media marketing of e-cigarette, we did not explore explicitly whether there were different perceptions of e-cigarette exposure in social media between boys and girls in the results. We acknowledge that this is an interesting factor to examine and would like to explore this in future research. We have amended the manuscript as follows:

Line 405: Future research could explore if there are different perceptions of e-cigarette exposure in social media between females and males. As females are one of the target groups for e-cigarette use

[66, 67], it can reflect the marketing situation of e-cigarettes for females on social media, which deserves further study.

References

1. Scottish Government. The Scottish Index of Multiple Deprivation 2020 2020 [Available from: <https://www.gov.scot/collections/scottish-index-of-multiple-deprivation-2020/>].
2. Smith MJ, Buckton C, Patterson C, Hilton S. User-generated content and influencer marketing involving e-cigarettes on social media: a scoping review and content analysis of YouTube and Instagram. *BMC Public Health*. 2023;23(1):530.
3. Braun V, Clarke V. Thematic analysis. *APA handbook of research methods in psychology, Vol 2: Research designs: Quantitative, qualitative, neuropsychological, and biological*. APA handbooks in psychology®.DOI: 10.1037/13620-004. Washington, D.C., USA: American Psychological Association; 2012. p. 57-71.

VERSION 2 – REVIEW

REVIEWER	Zhu, Jingfen Shanghai Jiao Tong University, School of Public Health
REVIEW RETURNED	12-Apr-2023
GENERAL COMMENTS	The authors have answered the proposed questions, and have revised them accordingly. It is hoped that the study will help to draw attention to the marketing of e-cigarettes in social media, so as to reduce teenagers' exposure to e-cigarettes on social media in Scotland .